# Evaluating the Distribution of Perfluoroalkyl Substances in Rice Paddy Lysimeter with an Andosol

**DOI:** 10.3390/ijerph191610379

**Published:** 2022-08-20

**Authors:** Heesoo Eun, Eriko Yamazaki, Yu Pan, Sachi Taniyasu, Kosuke Noborio, Nobuyoshi Yamashita

**Affiliations:** 1Research Center for Advanced Analysis, National Agriculture and Food Research Organization (NARO), 3-1-3 Kannondai, Tsukuba 305-8604, Ibaraki, Japan; 2National Institute of Advanced Industrial Science and Technology (AIST), 16-1 Onogawa, Tsukuba 305-8569, Ibaraki, Japan; 3Institute of Medicinal Plants, Yunnan Academy of Agricultural Sciences, Kunming 650200, China; 4Department of Agriculture, Meiji University, 1-1-1 Higashi-Mita, Kawasaki 214-8571, Kanagawa, Japan

**Keywords:** andosols, shorter chain PFAS, longer chain PFAS, lysimeter

## Abstract

The properties of potential emerging persistent contaminants, perfluoroalkyl substances (PFAS), in an andosol rice paddy lysimeter were analyzed to determine their mobility and leaching behavior regarding carbon chain length and functional groups. For this purpose, simulated contaminated water (ΣPFAS = 1,185,719 ng/L) was used in the lysimeter. The results showed that PFAS distribution in the paddy soil lysimeter was influenced by the migration of these substances into irrigation water and their adsorption into the soil. PFHxS (C6) and PFOS (C8), which are the main components of the simulated contaminated water, were mostly captured in the soil layers of the low-humic andosol layer (0–35 cm). PFAS distribution may depend on soil properties, such as total carbon (TC) content. Compared with perfluoroalkane sulfonic acids (PFSAs), the distribution of perfluoroalkyl carboxylic acids (PFCAs) in soil showed significant variation. The remaining PFCAs were distributed across all layers of the lysimeter, except for the longer-chain PFCAs. Moreover, the PFSA distribution was directly correlated with the carbon chain number, whereby longer- and shorter-chain PFSAs accumulated in the top and bottom soil layers, respectively. This study provides detailed information on the distribution, leaching, uptake, and accumulation of individual PFAS in andosol paddy fields in Japan.

## 1. Introduction

Perfluorooctane sulfonic acid (PFOS) and perfluorooctanoic acid (PFOA) are well-known perfluoroalkyl substances (PFAS). PFAS contain strong carbon–fluorine bonds, are hydrophobic and lipophobic, and dissipate extreme heat [1]. These characteristics have resulted in the widespread use of PFAS in adhesives, coatings, and firefighting foams [2]. However, PFAS have received increasing global attention because of their persistence and toxicity in the natural environment, posing a severe threat to human health [3,4,5]. As a result, PFAS were included in Annex B of the Stockholm Convention on Persistent Organic Pollutants (POPs) in 2009 [6]. Consequently, manufacturing and importing PFAS have been prohibited, and the use of PFOS and the associated salts for non-essential purposes has been prohibited in Japan after they were designated as Class 1 Specified Chemical Substances by the Chemical Substance Control Law (effective from May 2010), substantially limiting the manufacture and import of these substances. Moreover, this law elevates the importance of water quality analysis and management (No. 0330 issued by Raw Food on 30 March 2020) and increases the target value. The sum of the concentrations of these two substances was 0.00005 mg/L (provisional 50 ng/L).

Water and food consumption are generally considered the two primary sources of PFAS in humans, although air and air-suspended dust, food packaging, and cookware also contribute to the overall PFAS load in the human body [7]. Moreover, fish are considered one of the most important sources of PFAS exposure in humans [8,9]. As agricultural products are considered a less significant source of exposure to PFAS, few studies have focused on these substances in the agricultural environment, and the knowledge of PFAS in agricultural products and cultivation environments is limited.

Among agricultural products, rice is the staple food of more than half of the world’s population, with more than 3.5 billion people depending on this crop for more than 20% of their daily calorie requirements [10]. Asia accounts for 90% of global rice consumption, and paddy cultivation is the most prevalent method of rice farming in the East Asian region. The soil is prone to the uptake of hazardous chemicals, particularly PFAS, including PROS and PFOA, because of the flooded conditions required for paddy rice farming. However, the extent of research on PFAS in the East Asian rice paddies is limited [11,12,13], with most studies focusing on more accessible geographical locations. In addition, approximately 47% of Japan’s agricultural land is dominated by andosols (volcanic ash soil), which are characterized by a high carbon content. Globally, andosols are rare, and their distribution corresponds to less than 1% of the world’s total land area. Although previous studies have investigated the accumulation and distribution of PFAS based on lysimeters [14,15], to the best of our knowledge, this is the first time that PFAS in andosols are analyzed. Furthermore, because shorter-chain PFAS are more water-soluble than longer-chain PFAS [16,17], their increased mobility in rice paddy fields enhances the uptake of these harmful substances by the soil and promotes leaching into groundwater systems. In this study, the characteristics of PFAS in a paddy lysimeter were examined in terms of PFAS mobility, leaching behavior, and relationship to chain lengths and functional groups. Determining the mobilization potential of various PFAS in paddy fields is essential for understanding the retention and release of these POPs in agro- and aquatic environments.

## 2. Materials and Methods

### 2.1. Chemicals and Reagents

PFAC-MXB, a native PFAS mixture standard solution, was purchased from Wellington Laboratories Inc. (Guelph, ON, Canada). Perfluorobutane sulfonic acid (PFBS), perfluorohexanesulfonic acid (PFHxS), perfluorooctanesulfonic acid (PFOS), perfluorodecanesulfonic acid (PFDS), perfluorobutanoic acid (PFBA), perfluoropentanoic acid (PFPeA), perfluorohexanoic acid (PFHxA), perfluoroheptanoic acid (PFHpA), perfluorooctanoic acid (PFOA), perfluorononanoic acid (PFNA), perfluorodecanoic acid (PFDA), perfluoroundecanoic acid (PFUnDA), perfluorodecanoic acid (PFDoDA), perfluorotridecanoic acid (PFTrDA), perfluorotetradecanoic acid (PFTeDA), perfluorohexadecanoic acid (PFHxDA), perfluorooctadecanoic acid (PFOcDA), individual native standards, perfluorooctane sulfonamide (FOSA), N-ethyl perfluorooctane sulfonamide (*N*-EtFOSA), and N-ethyl perfluorooctane sulfonamide acetic acid (*N*-EtFOSAA) were procured from the same source.

The mass-labeled standard mixtures MPFAC-MXA, consisting of ^18^O_2_-PFHxS, ^13^C_4_-PFOS, ^13^C_4_-PFBA, ^13^C_2_-PFHxA, ^13^C_4_-PFOA, ^13^C_5_-PFNA, ^13^C_2_-PFDA, ^13^C_2_-PFUnDA, and ^13^C_2_-PFDoDA, and individual mass-labeled standards, ^13^C_3_-PFBS and ^13^C_5_-PFPeA, were purchased from Wellington Laboratories Inc. (Guelph, ON, Canada). All compounds had >98% purity.

Oasis^®^ weak anion exchange (WAX; 6 cc, 150 mg, 30 μm) solid-phase extraction (SPE) cartridges (Waters Corp., Milford, MA, USA), Supelclean^TM^ ENVI-Carb^TM^ (100 mg, 1 mL) cartridges (Sigma-Aldrich Corp., St. Louis, MO, USA), ammonium acetate (97%; Wako Pure Chemical Industries Ltd., Osaka, Japan), ammonium solution (25%; Wako Pure Chemical Industries Ltd., Osaka, Japan), acetic acid (>99.7%; guaranteed reagent; Wako Pure Chemical Industries Ltd., Osaka, Japan), ultrapure water (prepared with a Milli-Q^®^ gradient system; Millipore Co., Bedford, MA, USA), and methanol (>99.8%; pesticide residue-PCB analytical grade; Wako Pure Chemical Industries Ltd., Osaka, Japan) were used in all experiments and instrumental analyses.

### 2.2. Lysimeter Experiment

A rice paddy lysimeter with a square-shaped surface area of 4 m^2^ (2 × 2 m) and a depth of 1.8 m located at Meiji University, Japan, was the measuring device used in this study. The soil of the lysimeter consisted of a low-humic andosol volcanic ash soil from the surface to a depth of 35 cm and a Kanto loam below it (Figure 1). The lysimeter was equipped with a clear roof to prevent wet deposition and rainfall. The lysimeter experiment included the collection of leachate and pre- and post-experimental soil core samples. After the completion of puddling, 16.7 L of simulated contaminated water (mixture of river water and industrial wastewater) containing PFAS was applied to the lysimeter. Non-drinking tap water was used as irrigation water to maintain the water level of the paddy. The lysimeter water (leachate) was collected through seven drainpipes with depths of 5, 40, 75, 110, 145, 180, and 240 cm from May 2015 to November 2015, including the cultivation period of rice.

### 2.3. PFAS Analysis

Approximately 100 mL of the leachate sample spiked with 1 ng of mass-labelled PFAS was extracted using an Oasis^®^ WAX SPE cartridge. The leachate samples were analyzed following the modified international standard method [18], as described elsewhere. Briefly, the leachate sample was loaded onto an Oasis^®^ WAX SPE cartridge that had been pre-conditioned with 4 mL of 0.1% ammonium in methanol, methanol, and Milli-Q^®^ water. After washing the cartridge with 5 mL of Milli-Q^®^ water followed by 4 mL of 25 mM acetate buffer, the cartridge was centrifuged at 3000 rpm for 2 min. The target analytes were then eluted with 4 mL of methanol, followed by 0.1% ammonium in methanol. The eluate was passed through an Envi-Carb^TM^ cartridge, pre-conditioned with 3 mL of methanol for further cleanup. The eluate was concentrated to 1 mL under a gentle stream of nitrogen gas and transferred to a polypropylene vial for instrumental analysis.

A soil sample (5 g) spiked with 1 ng of mass-labeled PFAS was extracted with 20 mL of methanol and 4 mL of 50 mM ammonium acetate by mechanical shaking at 200 rpm for 30 min. The sample was then centrifuged at 3000 rpm for 30 min, and the supernatant was transferred to a new polypropylene (PP) tube. This extraction procedure was repeated twice. After extraction, the extract was diluted with 100 mL of Milli-Q^®^ water in a new PP bottle and extracted using an Oasis^®^ WAX SPE cartridge. The WAX SPE procedure was the same as that described above for the water samples.

The target analytes were separated and quantified using an Agilent 1100 liquid chromatograph (Agilent Technologies, Santa Clara, CA, USA) interfaced with a Micromass Quatro Ultima Pt mass spectrometer (Waters Corporation, Milford, MA, USA) in the electrospray negative ionization mode. The extracts were injected into a Betasil^®^ C18 column (2.1 mm i.d. × 50 mm length, 5 µm particle size; Thermo Fisher Scientific, Waltham, MA, USA) with a 2 mM ammonium acetate and methanol mixture as the mobile phase. The gradient consisting of 10% methanol was increased to 100% over 10 min and maintained for 14 min before reverting to the original conditions. The flow rate was 0.3 mL/min. The ion exchange type analytical column (RSpak JJ-50 2D; 2.0 mm i.d. × 150 mm length; Shodex, Showa Denko K.K., Kawasaki, Japan) with a 50 mM ammonium acetate and methanol mixture as the mobile phase at a flow rate of 0.3 mL/min was used especially for the conformation of shorter chain PFAS results. A 10 µL sample extract aliquot was injected into both analytical columns. The variation in PFAS concentration between the two columns was <10%. The desolvation gas flow rate and temperature were maintained at 610 L/h and 450 °C, respectively. The collision energies, cone voltages, and detailed MS/MS parameters for the instrument were optimized for individual analytes and were similar to those reported elsewhere [18].

### 2.4. Quality Assurance and Quality Control

Procedural blanks and recoveries were analyzed for every batch of samples and treated using the same procedure used for the real samples. The mass-labeled PFAS was spiked in each sample before extraction to check the overall recovery of the target analytes. The instrumental limits of quantitation (I-LOQs) of each target analyte were defined as the minimum measured concentrations in the calibration standards, resulting in a signal-to-noise ratio of ≥10. The concentrations found in procedural blanks were all below their corresponding method quantification limits (MQLs), ranging from <20–<500 pg/L for irrigation tap water, <0.1–<4 ng/L for simulated contaminated water, <0.02–<20 ng/L for leachate, and <1–<117 pg/g-dry weight for soil samples. Procedural recoveries ranged from 79 to 97% (*n* = 14, S.D. 6–13%). The mass-labeled PFAS recoveries in each matrix ranged from 90–100% for irrigation tap water, 62–107% for simulated contaminated water, 89–96% for leachate, and 72–83% for soil samples. The concentrations were not corrected by mass-labeled PFAS recoveries.

## 3. Results and Discussion

Rice is mainly cultivated in the paddy fields of East Asia. Because rice is cultivated in a water-soaked environment most of the time (approximately 50–60 d) during its growth process, the vertical movement of PFAS in water to groundwater is very important. PFAS is considered to penetrate groundwater depending on the soil properties (for example, total carbon content), as the water solubility largely depends on the length and functional group of PFAS. Thus, the penetration of PFAS from agricultural land soil into groundwater is an important issue from the viewpoint of ensuring the safety of drinking water. Table 1 shows the concentrations of PFAS in the simulated contaminated water and non-drinking tap water used for irrigation. For the simulated contaminated water, PFHxS and PFOS were the predominant PFAS, with concentrations of 702,885 and 395,909 ng/L, respectively. The sum of perfluoroalkane sulfonic acid (PFSAs), FOSA, and FOSAA concentrations (ΣPFSAs + FOSA + FOSAA) and perfluoroalkyl carboxylic acid (PFCAs) concentrations (ΣPFCAs) accounted for 95% and 5% of the total PFAS concentration (ΣPFAS), respectively. In particular, the target compounds of risk assessment [19], PFHxS (59%), and PFOS (33%) had the highest contribution to the composition profile of ΣPFAS.

In contrast, the ΣPFSAs + FOSA + FOSAA (3512 ng/L and ΣPFCAs (4121 ng/L) in tap water for irrigation accounted for 46% and 54%, respectively. PFHxS, PFOS, PFOA, and PFBA were the dominant PFAS, accounting for 23%, 16%, 15%, and 11% of the ΣPFAS, respectively. The PFAS concentrations in the tap water used for irrigation were comparable to those collected from other locations in Japan, as reported previously [20,21].

The depth distribution of PFAS in the pre- and post-experimental soil cores of the lysimeter is shown in Figure 2. After simulated contaminated water was applied to the lysimeter, it was found that PFSAs remained from the surface layer to the 45 cm depth layer depending on the depth. In the pre-soil core, the concentration of PFCA was 5–6 times that of PFSAs. Due to irrigation containing PFAS, the balance of PFAS in the soil of the lysimeter changed considerably. PFSAs changed into soil at a much higher concentration than PFCAs did. In particular, PFHxS (C6) and PFOS (C8), which are the main components of the simulated contaminated water, were mostly captured in the soil layers of the low-humic andosol layer (0–35 cm). The concentrations of PFHxS and PFOS increased because the high concentrations of PFHxS and PFOS were primarily derived from irrigation containing PFAS. The residual PFSAs in paddy soils depend on the PFSAs components of irrigation water. In addition, a longer chain of PFSA, FOSA, and FOSAA also appeared to be captured in the surface layer, while shorter chains (C ≤ 4), such as PFBA and PFPeA, showed insignificant fluctuations. Furthermore, PFSAs are more easily affected by the carbon chain number than PFCAs. When the carbon chain number was greater than 10, the concentration of carboxylic acids also decreased in the middle or bottom layer. Compared with the soil core, carboxylic acids tend to remain in water rather than in soil, which may be associated with their functional groups. For example, the water solubility of PFHxS is approximately 1.49 × 10^−6^ to 5.69 × 10^−1^ mol/L. Moreover, the TC content of the upper soil layer in this study was higher (approximately 4%) than that of the lower layers (2–3%). Thus, PFHxS was strongly absorbed by the relatively high TC soil particles in the upper layer. Indeed, the middle- and longer-chain PFSAs remained in the upper soil layer, owing to their water solubility and the TC properties of the soil. In contrast, because the shorter-chain PFSAs have relatively higher water solubilities, the migration of these substances is primarily controlled by the movement of soil water and not the TC of the soil.

It is widely known that andosols have a high total carbon content. Among them, low-humic andosol is soil with a higher total carbon content than general andosols. Paddy fields for rice cultivation are generally fertilized by mixing rice straw and livestock manure as organic fertilizers annually and plunging into the soil for soil management by farmers. At that time, the surface paddy soil (approximately 30 cm deep) was stirred several times to make it uniform using a cultivator machine. Therefore, the total carbon content from the surface layer of the paddy soil to a depth of approximately 30 cm was higher than that of the soil layer below. In other words, although the surface layer of the paddy in andosol originally had a high total carbon content, the total carbon content in the surface layer (approximately 30 cm depth) increased every year because of the additional input of organic fertilizer. Thereby, PFSAs may be strongly adsorbed on the upper layer of soil with high carbon content. Furthermore, because PFSAs have a lower water solubility than PFCAs, they are easily adsorbed into soil; thus, they remain on the upper-layered soil for a long time. Stahl et al. [15] reported that PFOA was transported rapidly by water passing through the soil, whereas PFOS traveled much more slowly in leachate in a lysimeter experiment. Therefore, the migration behavior of PFAS in paddy fields is associated with their functional groups, such as carboxylic acids or sulfonic acids.

After the simulated contaminated water was applied to the paddy (on June 21), the PFAS concentration (mainly for PFHxS and PFOS) in the leachate of the 5 cm layer significantly increased, and then dramatically decreased after 2 d (Figure 3 and Figure 4). Compared with the results of the soil core, PFAS seemed to be mainly transferred to or remaining in the soil.

Contrastingly, in the 40, 75, and 110 cm layers below it, PFAS concentration rose steadily, peaked from July 23 to August, and showed a declining trend. Unfortunately, PFAS values at the 110, 145, and 180 cm layers from June 18 to July 7 were not obtained. In this study, we focused on the movement of high concentrations of PFHxS and PFOS from the bottom of simulated contaminated water. Most of the PFHxS and PFOS were adsorbed on the low-humic andosol layer (0–35 cm), but it was revealed that the lower part penetrated upward over approximately four months. In contrast, the 145 cm and 180 cm layers showed a sharp decreasing tendency after reaching the peak of PFHxS and PFOS concentrations in July. It was found that even PFHxS and PFOS continued to infiltrate and shift downward over four months or more. Overall, the results indicate that even low-humic andosol and andosol with abundant carbon content may migrate to groundwater while passing through the theoretical plate number of soil according to the movement of water directly underneath.

## 4. Conclusions

To the best of our knowledge, this is the first study of the PFAS distribution in paddy soils containing andosols in Japan. The results showed that the distribution of PFAS in paddy soils was influenced by their migration in water and adsorption to the soil. Among them, PFHxS (C6) and PFOS (C8), which are the main components of the simulated contaminated water (ΣPFAS = 1,185,719 ng/L), were mostly captured in the soil layers of the low-humic andosol layer (0–35 cm). The PFAS distribution may depend on soil properties, such as the TC content of the soil. In particular, PFSAs may be strongly adsorbed on the upper layer of soil with high carbon content. Because PFSAs have a lower water solubility than PFCAs, they are easily adsorbed into soil; thus, they remain on the upper-layered soil for a long time. However, it was difficult to identify the separate effects of diffusion in the water and adsorption–desorption to the soil due to differences in the individual chemical properties of PFAS. This includes water solubility and carbon content varying in each soil layer analyzed in the lysimeter. Interestingly, PFSAs are more easily affected by the carbon chain number than PFCAs. Further studies are required to understand the distribution and leaching of PFAS in paddy fields to mitigate the adverse effects of these substances on human health.

## Figures and Tables

**Figure 1 ijerph-19-10379-f001:**
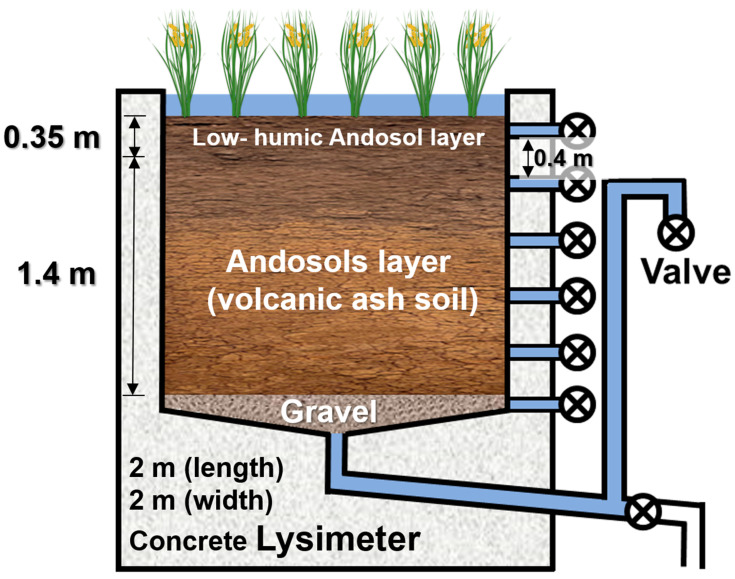
Schematic figure of lysimeter used for paddy cultivation in this study.

**Figure 2 ijerph-19-10379-f002:**
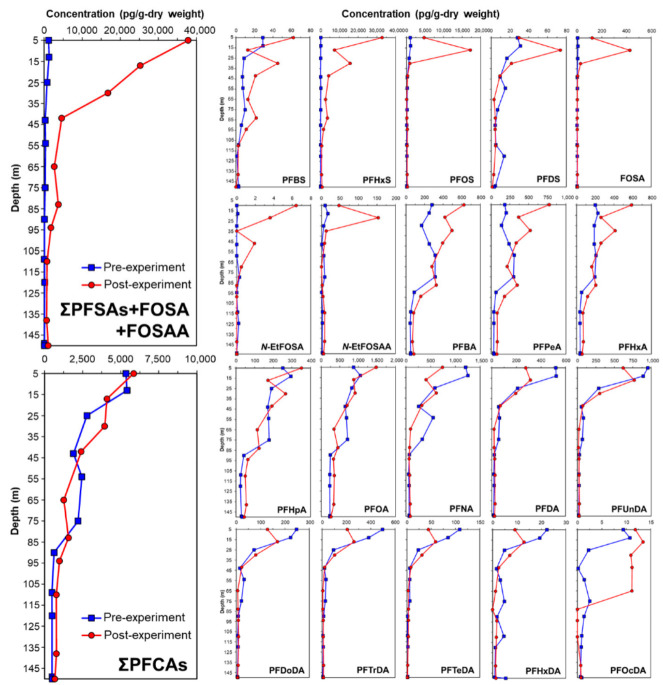
Concentrations (pg/g-dry weight) of PFAS in pre- and post-experimental lysimeter soil core.

**Figure 3 ijerph-19-10379-f003:**
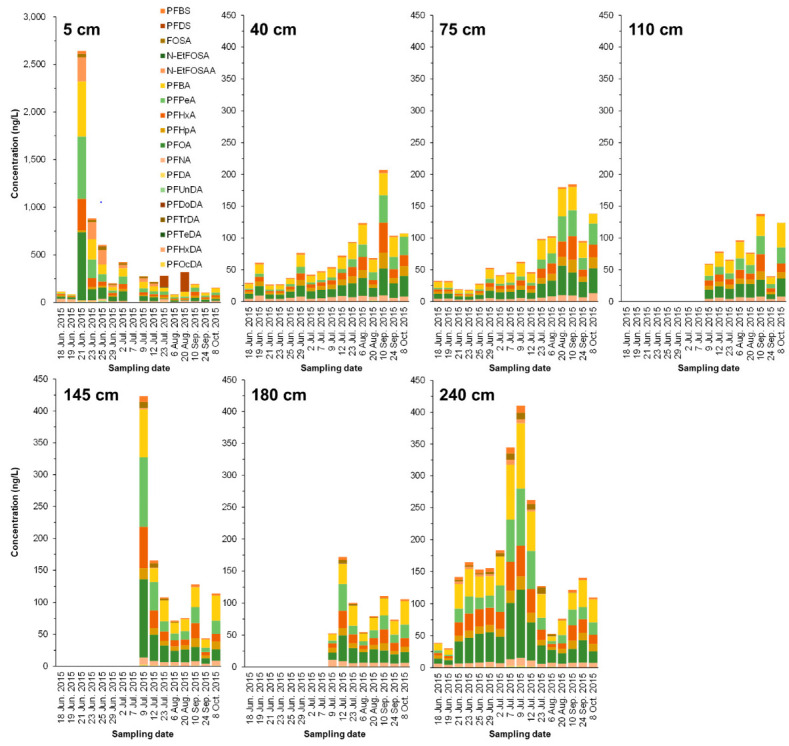
Time-course change of PFAS concentrations (ng/L) in lysimeter leachate collected from different depths (5, 40, 75, 110, 145, 180, and 240 cm). The leachate samples could not be obtained for all sampling dates.

**Figure 4 ijerph-19-10379-f004:**
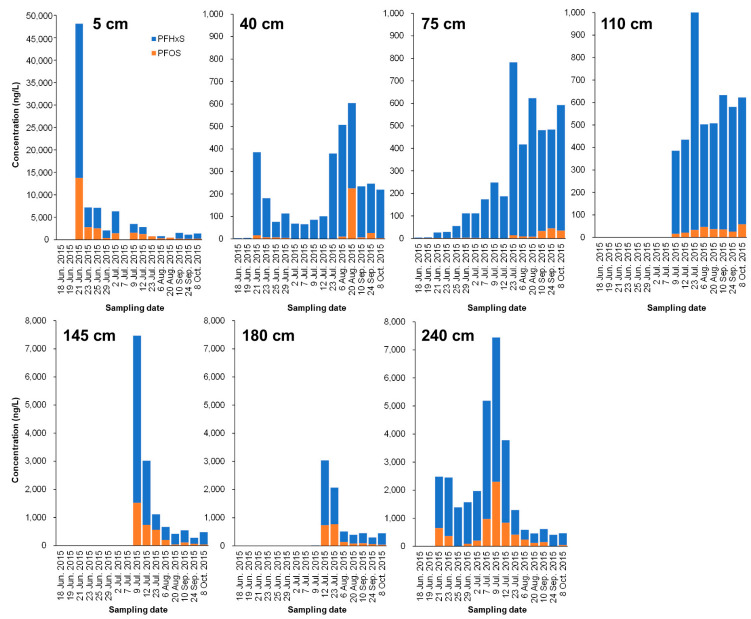
Time-course change of PFHxS and PFOS concentrations (ng/L) in lysimeter leachate collected from different depths (5, 40, 75, 110, 145, 180, and 240 cm). The leachate samples could not be obtained for all sampling dates.

**Table 1 ijerph-19-10379-t001:** Concentrations of PFAS in simulated contaminated water (ng/L) and tap water for irrigation (pg/L) used for the lysimeter experiment.

Group	Analyte	IUPAC Name	Formula	Simulated Contaminated Water	Tap Water for Irrigation
				(ng/L)	(pg/L)
PFSA	PFBS	1,1,2,2,3,3,4,4,4-Nonafluorobutane-1-sulfonic acid	C_4_HF_9_O_3_S	1484	352
	PFHxS	1,1,2,2,3,3,4,4,5,5,6,6,6-Tridecafluorohexane-1-sulfonic acid	C_6_HF_13_O_3_S	702,885	1721
	PFOS	1,1,2,2,3,3,4,4,5,5,6,6,7,7,8,8,8-Heptadecafluorooctane-1-sulfonic acid	C_8_HF_17_O_3_S	395,909	1240
	PFDS	1,1,2,2,3,3,4,4,5,5,6,6,7,7,8,8,9,9,10,10,10-Henicosafluorodecane-1-sulfonic acid	C_10_HF_21_O_3_S	364	33
FOSA	FOSA	1,1,2,2,3,3,4,4,5,5,6,6,7,7,8,8,8-Heptadecafluoro-1-octanesulfonamide	C_8_H_2_F_17_NO_2_S	1257	48
	*N*-EtFOSA	N-Ethyl-1,1,2,2,3,3,4,4,5,5,6,6,7,7,8,8,8-heptadecafluorooctane-1-sulfonamide	C_10_H_6_F_17_NO_2_S	4234	<100
FOSAA	*N*-MeFOSAA	2-[1,1,2,2,3,3,4,4,5,5,6,6,7,7,8,8,8-Heptadecafluorooctylsulfonyl(methyl)amino]acetic acid	C_11_H_6_F_17_NO_4_S	19,468	118
PFCA	PFBA	2,2,3,3,4,4,4-Heptafluorobutanoic acid	C_4_HF_7_O_2_	14,754	858
	PFPeA	2,2,3,3,4,4,5,5,5-Nonafluoropentanoic acid	C_5_HF_9_O_2_	15,100	391
	PFHxA	2,2,3,3,4,4,5,5,6,6,6-Undecafluorohexanoic acid	C_6_HF_11_O_2_	7873	647
	PFHpA	2,2,3,3,4,4,5,5,6,6,7,7,7-Tridecafluoroheptanoic acid	C_7_HF_13_O_2_	3198	528
	PFOA	2,2,3,3,4,4,5,5,6,6,7,7,8,8,8-Pentadecafluorooctanoic acid	C_8_HF_15_O_2_	19,156	1166
	PFNA	2,2,3,3,4,4,5,5,6,6,7,7,8,8,9,9,9-Heptadecafluorononanoic acid	C_9_HF_17_O_2_	1.5	498
	PFDA	2,2,3,3,4,4,5,5,6,6,7,7,8,8,9,9,10,10,10-Nonadecafluorodecanoic acid	C_10_HF_19_O_2_	27	34
	PFUnDA	2,2,3,3,4,4,5,5,6,6,7,7,8,8,9,9,10,10,	C_11_HF_21_O_2_	5.4	<100
		11,11,11-Henicosafluoroundecanoic acid			
	PFDoDA	2,2,3,3,4,4,5,5,6,6,7,7,8,8,9,9,10,10,	C_12_HF_23_O_2_	2.6	<100
		11,11,12,12,12-Tricosafluorododecanoic acid			
	PFTrDA	2,2,3,3,4,4,5,5,6,6,7,7,8,8,9,9,10,10,	C_13_HF_25_O_2_	0.59	<100
		11,11,12,12,13,13,13-Pentacosafluorotridecanoic acid			
	PFTeDA	2,2,3,3,4,4,5,5,6,6,7,7,8,8,9,9,10,10,	C_14_HF_27_O_2_	0.26	<100
		11,11,12,12,13,13,14,14,14-Heptacosafluorotetradecanoic acid			
	PFHxDA	2,2,3,3,4,4,5,5,6,6,7,7,8,8,9,9,10,10,	C_16_HF_31_O_2_	0.038	<500
		11,11,12,12,13,13,14,14,15,15,16,16,			
		16-Hentriacontafluorohexadecanoic acid			
	PFOcDA	2,2,3,3,4,4,5,5,6,6,7,7,8,8,9,9,10,10,	C_18_HF_35_O_2_	<0.1	<20
		11,11,12,12,13,13,14,14,15,15,16,16,			
		17,17,18,18,18-Pentatriacontafluorooctadecanoic acid			
	ΣPFSAs +FOSA+ FOSAA			1,125,601	3512
	ΣPFCAs			60,118	4121
	ΣPFAS			1,185,719	7634

## Data Availability

Not applicable.

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
