# Peer review of "Evaluating the Distribution of Perfluoroalkyl Substances in Rice Paddy Lysimeter with an Andosol"

_ijerph, 2022, doi:10.3390/ijerph191610379_

Round 1
Reviewer 1 Report
Title: Evaluating the Distribution of Perfluoroalkyl Substances in Rice Paddy Lysimeter with an Andosol
1: Abstract should be concise and quantitative as per results.
2: Nomenclature list should be given at the starting the introduction?
3: Authors showed the study of 2015 and now 2022? “The lysimeter water (leachate) was collected through 7 drain 111 pipes with depths of 5, 40, 75, 110, 145, 180, and 240 cm from May 2015 to November 2015, including the 112 cultivation period of rice”
4: Figure 3 should be split two parts so that visibility will be better?
5: Discussion section need to be improved through recent studies? Generally, authors used old citation that will remove and cite new studies?
6: Units used in study will follow same pattern or SI or FPS?
7: conclusion should be focus and quantitative as per results.
8: Referencing as per journal guidelines.
Author Response
1: Abstract should be concise and quantitative as per results.
Response: We thank you for the insightful comments, which have helped us to substantially improve our manuscript. According to the suggestions, we have thoroughly revised our manuscript.
We have substantially revised the abstract per your comment.
L14–21
The properties of potential emerging persistent contaminants, perfluoroalkyl substances (PFAS), in an Andosol rice paddy lysimeter were analyzed to determine their mobility and leaching behavior based on their carbon chain length and functional groups. For this purpose, simulated contaminated water (ΣPFAS= 1185719 ng/L) was used in the lysimeter. The results showed that PFAS distribution in the paddy soil lysimeter was influenced by the migration of these substances into irrigation water and their adsorption on the soil. PFHxS (C6) and PFOS (C8), which are the main components of the simulated contaminated water, were mostly captured in the soil layers of the low-humic andosol layer (0–35 cm).
2: Nomenclature list should be given at the starting the introduction?
Response: We thank you for the comment. The abbreviation of the target compounds of this study has been presented in the CHEMICALS AND REAGENTS section, and more details are presented in Table 1.
L76–92
PFAC-MXB, a native PFAS mixture standard solution, was purchased from Wel-lington Laboratories Inc. (Guelph, ON, Canada). Perfluorobutane sulfonic acid (PFBS), perfluorohexanesulfonic acid (PFHxS), perfluorooctanesulfonic acid (PFOS), per-fluorodecanesulfonic acid (PFDS), perfluorobutanoic acid (PFBA), perfluoropentanoic acid (PFPeA), perfluorohexanoic acid (PFHxA), perfluoroheptanoic acid (PFHpA), per-fluorooctanoic acid (PFOA), perfluorononanoic acid (PFNA), perfluorodecanoic acid (PFDA), perfluoroundecanoic acid (PFUnDA), perfluorodecanoic acid (PFDoDA), per-fluorotridecanoic acid (PFTrDA), perfluorotetradecanoic acid (PFTeDA), perfluorohex-adecanoic acid (PFHxDA), perfluorooctadecanoic acid (PFOcDA), individual native standards, perfluorooctane sulfonamide (FOSA), N-ethyl perfluorooctane sulfonamide (N-EtFOSA), and N-ethyl perfluorooctane sulfonamide acetic acid (N-EtFOSAA) were procured from the same source.
The mass-labeled standard mixtures MPFAC-MXA, consisting of 18O2-PFHxS, 13C4-PFOS, 13C4-PFBA, 13C2-PFHxA, 13C4-PFOA, 13C5-PFNA, 13C2-PFDA, 13C2-PFUnDA, and 13C2-PFDoDA, and individual mass-labeled standards 13C3-PFBS and 13C5-PFPeA were purchased from Wellington Laboratories Inc. (Guelph, ON, Canada). All compounds had >98% purity.
3: Authors showed the study of 2015 and now 2022? “The lysimeter water (leachate) was collected through 7 drain 111 pipes with depths of 5, 40, 75, 110, 145, 180, and 240 cm from May 2015 to November 2015, including the 112 cultivation period of rice”
Response: In this study, we artificially added high-concentration PFAS cultivation water into the lysimeter and observed the movement. Unfortunately, follow-up was not possible because the lysimeter was subsequently used in another study. As the test lysimeter is in a university facility, it is used for various experiments.
4: Figure 3 should be split two parts so that visibility will be better?
Response: We thank you for the suggestion. Figure 3 has been split into two figures accordingly.
5: Discussion section need to be improved through recent studies? Generally, authors used old citation that will remove and cite new studies?
Response: Per your comment, we have replaced old citations with new studies where possible.
6: Units used in study will follow same pattern or SI or FPS?
Response: We have used SI units throughout the manuscript.
7: conclusion should be focus and quantitative as per results.
Response: Per your comment, we have revised the conclusion as follows.
L262–269
The results showed that the distribution of PFAS in paddy soils was influenced by their migration in water and adsorption to the soil. Among them, PFHxS (C6) and PFOS (C8), which are the main components of the simulated contaminated water (ΣPFAS= 1185719 ng/L), were mostly captured in the soil layers of the low-humic andosol layer (0–35 cm). The PFAS distribution may depend on soil properties such as the TC content of the soil. In particular, PFSAs may be strongly adsorbed on the upper layer of soil with high carbon content. Because PFSAs have a lower water solubility than PFCAs, they are easily adsorbed into soil; thus, they remain on the upper-layered soil for a long time.
8: Referencing as per journal guidelines.
Response: The references have been formatted per the journal guidelines.

Reviewer 2 Report
In the study, the authors analyzed PFAS substances in an Andosol rice paddy lysimeter to determine/discuss their mobility and leaching behavior regarding carbon chain length and functional groups. The study is well presented and I have some suggestions:
Introduction: remove the last paragraph L77-L79.
Materials and Methods: Include the chemical structures for all molecules.
Results and Discussion: It is important to show the error values for all measurements.
Author Response
In the study, the authors analyzed PFAS substances in an Andosol rice paddy lysimeter to determine/discuss their mobility and leaching behavior regarding carbon chain length and functional groups. The study is well presented and I have some suggestions:
Response: We appreciate the positive comments and valuable suggestions. We have revised the manuscript accordingly. Our responses to the comments are provided below. 
Introduction: remove the last paragraph L77-L79.
Response: The paragraph has been deleted.
Materials and Methods: Include the chemical structures for all molecules.
Response: Due to the limitation on the length of the manuscript set by the journal and the large number of PFAS in the study, it is difficult to show the chemical structures of all molecules. Instead, we have added the IUPAC name and Chemical Abstract Services Registry Number in Table 1.
Results and Discussion: It is important to show the error values for all measurements.
Response: We thank you for your valuable suggestions.
We have included the error values in the QUALITY assurance and QUALITY CONTROL section.

Reviewer 3 Report
The manuscript " Evaluating the distribution of...." reports a very interesting study on perfluoroalkyl substances in an andosol paddy lysimeter. The results that the authors report showed that the distribution of PFAS in the soil lysimeter of the paddy field was influenced by the migration of these substances into the irrigation water and their absorption in the soil and the total carbon content of the soil. Compared to perfluoroalkane sulfonic acids, the distribution of perfluoroalkyl carboxylic acids in the soil showed significant variations. The remainig PFCAs were distribuited over all layers of the lysimeter except for the longer chain PFCAs. Furthemore the distribution of PFSA was directly related to the carbon chain number, whereby longer chain PFSA accumulated in the upper soil layer and shorter chain PFSA poured into the lower soil layers. However it was not possible to distinguish the individual effects of water diffusion and soil adsorption-desorption due tp the different chemical properties including water solubility and carbon content of the PFAS analyzed in each soil layer. In my opinion the work for its significant even if not exhaustive content can be taken into considetation for the publication on Int. J. Environ. Res. Public Health.
Author Response
he manuscript " Evaluating the distribution of...." reports a very interesting study on perfluoroalkyl substances in an andosol paddy lysimeter. The results that the authors report showed that the distribution of PFAS in the soil lysimeter of the paddy field was influenced by the migration of these substances into the irrigation water and their absorption in the soil and the total carbon content of the soil. Compared to perfluoroalkane sulfonic acids, the distribution of perfluoroalkyl carboxylic acids in the soil showed significant variations. The remainig PFCAs were distribuited over all layers of the lysimeter except for the longer chain PFCAs. Furthemore the distribution of PFSA was directly related to the carbon chain number, whereby longer chain PFSA accumulated in the upper soil layer and shorter chain PFSA poured into the lower soil layers. However it was not possible to distinguish the individual effects of water diffusion and soil adsorption-desorption due tp the different chemical properties including water solubility and carbon content of the PFAS analyzed in each soil layer. In my opinion the work for its significant even if not exhaustive content can be taken into considetation for the publication on Int. J. Environ. Res. Public Health.
Response: We express our deep gratitude for your positive feedback regarding our manuscript.

Round 2
Reviewer 1 Report
Revised manuscript updated as per given comments.
Now this will be accepted for publication.